# Methodology for the Detection of Contaminated Training Datasets for Machine Learning-Based Network Intrusion-Detection Systems [note 1]

**DOI:** 10.3390/s24020479

**Published:** 2024-01-12

**Authors:** Joaquín Gaspar Medina-Arco, Roberto Magán-Carrión, Rafael Alejandro Rodríguez-Gómez, Pedro García-Teodoro

**Affiliations:** Network Engineering & Security Group (NESG), University of Granada, 18012 Granada, Spain; rmagan@ugr.es (R.M.-C.); rodgom@ugr.es (R.A.R.-G.); pgteodor@ugr.es (P.G.-T.)

**Keywords:** anomaly detection, NIDS, deep learning, autoencoders, methodology, real network datasets, data quality

## Abstract

With the significant increase in cyber-attacks and attempts to gain unauthorised access to systems and information, Network Intrusion-Detection Systems (NIDSs) have become essential detection tools. Anomaly-based systems use machine learning techniques to distinguish between normal and anomalous traffic. They do this by using training datasets that have been previously gathered and labelled, allowing them to learn to detect anomalies in future data. However, such datasets can be accidentally or deliberately contaminated, compromising the performance of NIDS. This has been the case of the UGR’16 dataset, in which, during the labelling process, botnet-type attacks were not identified in the subset intended for training. This paper addresses the mislabelling problem of real network traffic datasets by introducing a novel methodology that (i) allows analysing the quality of a network traffic dataset by identifying possible hidden or unidentified anomalies and (ii) selects the ideal subset of data to optimise the performance of the anomaly detection model even in the presence of hidden attacks erroneously labelled as normal network traffic. To this end, a two-step process that makes incremental use of the training dataset is proposed. Experiments conducted on the contaminated UGR’16 dataset in conjunction with the state-of-the-art NIDS, Kitsune, conclude with the feasibility of the approach to reveal observations of hidden botnet-based attacks on this dataset.

## 1. Introduction

Network Intrusion-Detection Systems (NIDSs) represent a primary cybersecurity mechanism for identifying potential attacks on a communication network. To accomplish this goal, they analyse the network traffic passing through the system, regardless of whether it is internally generated or originated from external entities targeting the network. Detecting intrusions allows network administrators to become aware of system vulnerabilities and to make quick decisions to abort or mitigate attacks. Additionally, NIDSs allow them to implement measures to strengthen the system in the future [1].

NIDSs can be categorised into various typologies based on two fundamental principles: architecture and techniques employed. Focusing on the architecture, NIDS can be classified as host-based, network-based, and collaborative approaches between different components. According to the detection technique, the classification may be signature-based, Stateful Protocol Analysis-based, or anomaly detection-based NIDSs [2].

Signature-based NIDSs possess a repository of network patterns representing prevalent network attacks. Their operating mode is to match the network sequence they examine with their knowledge base to detect potential attacks [3].

Alternatively, Stateful Protocol Analysis-based NIDSs rely on their comprehensive understanding of the monitored protocol. They analyse all interactions to identify a sequence of actions that might result in a vulnerability or insecurity [3].

In contrast, anomaly-detection-based NIDSs employ mechanisms to detect abnormal network traffic behaviour. These anomalous activities typically correspond to network traffic patterns that have a significantly low likelihood of occurring or are markedly misaligned with normal traffic. Acutely objective, anomaly detection allows for the handling of novel or previously unknown attacks (*zero days*). This is because such attacks generate traffic patterns that have not been found before, and this type of NIDS often relies on the use of machine learning techniques to carry out anomaly detection. When this approach is followed, the subjective evaluation of attacks is effectively circumvented.

Different strategies have been employed to detect anomalies in NIDS through various machine learning techniques [4,5], including statistical techniques like Principal Component Analysis (PCA) [6] or Markov models [7,8]; classification techniques like Artificial Neural Networks (ANNs) [9,10,11,12], Support Vector Machines (SVMs) [6], deep learning models [13,14] including Autoencoders [9,15], or Decision Trees including Random Forest [16]; and clustering like outlier detection [17]. Using these techniques requires a multi-perspective approach to tackling the problem, which can be categorised as supervised, semi-supervised, or unsupervised, depending on the specific technique chosen [18].

Regardless of the technique used for anomaly detection in NIDS, the underlying models must be trained to distinguish normal traffic from anomalous traffic. This training process utilises datasets comprising real, synthetic, or a combination of both network traffic. To be more concise,

Synthetic traffic datasets are created by generating traffic in a controlled environment that emulates a real-world setting. The generated traffic may include traffic related to known attacks, providing enough samples for machine learning models to competently identify and detect such anomalies. This enables the optimisation of the dataset regarding the size and balance between regular and irregular traffic samples. It also ensures the correct labelling of each observation as it has been intentionally and deliberately generated. Such observations can be, for instance, the traffic flows seen in the network. However, a potential issue is that it may not accurately reflect the network traffic patterns observed in a genuine environment.Real traffic datasets capture all network communications within a real productive environment. This implies access to the patterns of network traffic consumption and usage that take place in an actual scenario and potentially any cyber-attacks that may occur. Unlike synthetic datasets, real traffic samples may be biased or imbalanced, with the presence of anomalous traffic often being minimal or completely absent. It is necessary to carry out a subsequent process to assign a normality or attack label to each flow for its use in machine learning models during training phases.Composite datasets are the ones generated by combining real environment data and synthetic traffic to introduce attack patterns.

Regardless of the AI model used in a NIDS, the dataset’s labelling accuracy is crucial to maintaining high model performance. This principle applies equally to supervised and unsupervised learning. In supervised learning, labelling is necessary to enable models to learn how to identify anomalous traffic. In contrast, unsupervised learning generally assumes that the training dataset consists of normal traffic only and is, therefore, free of anomalies.

An example of such a dilemma can be observed in the present traffic dataset UGR’16, where irregularities such as botnet-type attacks were detected in the training set, which remained unlabelled for several months [19,20]. Consequently, the detection performance the authors claim when using this dataset can be called into question.

This work presents a methodology for detecting potentially hidden anomalies to prevent the mislabelling of a real traffic dataset. Mislabelling can occur accidentally or intentionally, as in the case of UGR’16, and can undermine or poison the artificial intelligence models trained on it. The primary aim of this framework is to enhance the reliability of dataset labelling, specifically within the training subset, by preventing unidentified anomalies from adversely affecting the efficacy of models trained on the data. This is achieved by providing a mechanism to increase confidence in the labelling. In addition, as a complement to this objective, the methodology can also be applied to select a subset of data from the full training set to optimise anomaly detection results.

The main contributions of this study are therefore as follows:The proposal of a methodology to identify concealed anomalies or contamination in real network traffic data.This technique allows for minimising the size of the training data set while maximising the efficiency of inference in artificial intelligence models.The methodology integrates Kitsune, a state-of-the-art NIDS, as a fundamental step to analyse the corrupted UGR’16 dataset, showcasing its efficacy.

### Motivation

Advances in communications, such as the Internet of Things (IoT), edge computing, or cloud computing, as well as the evolution of customer–supplier or employer–employee relationship models, are placing computer networks at the centre and making them the key element underpinning this ecosystem. As an unintended consequence, computer networks are also becoming the target of many cyber-attacks or the tools to carry them out. The revolution of IoT devices and their exponential growth is amplifying the nature of attacks, with IoT devices being used as instruments to carry out attacks with previously unknown capabilities [21], such as the well-known Mirai attack in 2016 [22].

One of the measures to deal with these possible security breaches is to detect the attacks in order to be able to act against them, using elements such as NIDSs, whose purpose is to determine whether the network traffic observed corresponds to an attack or not. The objective of NIDSs is therefore to be able to classify each traffic flow detected as benign or malicious in a binary manner [23]. Machine-learning-based techniques are commonly used to implement them [24].

Therefore, NIDSs play a crucial and increasingly important role, as they can be the first line of defence against cyber-attacks. Our work pursues the robustness of NIDSs through a methodology that allows for better training of the underlying AI models that perform the classification of network flows and ensures that the data used for such training is error-free. Our work is therefore in line with the current challenges:Cybersecurity and AI: The application of AI in the field of cybersecurity requires progress in its own protection because protecting the protectors is needed.Data quality: The generation of datasets, both real and synthetic, in the field of network traffic, is a complex process on which detection and prevention processes and tools depend on, so it is necessary to work on maximising their quality by reducing potential errors.

The article is structured as follows. Section 2 discusses related work on intrusion detection and datasets. In Section 3, methods and materials are introduced. Section 4 describes the proposed methodology, while Section 5 delves into the implementation of the methodology in one use case and the obtained results. In Section 6, those results are discussed, and, finally, Section 7 summarises the findings and suggests possible future research.

## 2. Related Works

### 2.1. Datasets for Network Security Purposes

To effectively train any AI model, especially those constituting NIDSs based on anomaly detection, a prerequisite is a comprehensive dataset. This dataset should encompass a sufficient number of samples that represent all the various classes or patterns, whether benign or malicious. This foundational dataset enables the model to learn and predict accurately during subsequent training phases. In the specific case of NIDSs, a large and correctly labelled dataset is assumed [23]. The quality of the trained models depends to some extent on the quality of the data on which they were trained [25], so it is important to make a thorough analysis of the typology of datasets available in the NIDS domain.

Before reviewing the different datasets available in the field of cybersecurity, it is necessary to define the criteria according to which these datasets will be analysed:Availability: Understood as free access (Public) to the dataset or, on the contrary, of reserved access, by means of payment or explicit request (Protected).Collected data: Some datasets collect traffic packet for each packet (e.g., PCAP files), others collect information associated with traffic flows between devices (e.g., NetFlow), and others extract features from the flows by combining them with data extracted from the packets.Labelling: This refers to whether each observation in the dataset has been identified as normal, anomalous, or even belonging to a known attack. Or, conversely, no labelling is available, in which case they are intended for unsupervised learning models.Type: The nature of a dataset may be synthetic, where the process and environment in which the dataset is generated are controlled, or it may be the result of capturing traffic in a real environment.Duration: Network traffic datasets consist of network traffic recorded over a specific time interval, which may range from hours to days, months, or even years.Size: the depth of the dataset in terms of the number of records or the physical size and their distribution across the different classes.Freshness: It is also important to consider the year in which the dataset was created, as the evolution of attacks and network usage patterns may not be reflected in older datasets, thus compromising their validity in addressing current issues.

A summary of the datasets analysed according to the characteristics described above is shown in Table 1.

#### 2.1.1. DARPA Datasets

Created by MIT’s Lincoln Laboratory, the DARPA datasets, with KDD datasets, are perhaps the most widely used in the field of intrusion-detection systems [35]. There are two versions, one created in 1998 and the other in 1999. Both collect synthetically generated network packets in controlled network environments simulating network traffic patterns previously observed in production environments. In the case of the 1998 version, the duration of the training subset is seven weeks of data, while in the 1999 version, the training subset consists of only three weeks of observations. In both cases, two weeks of observed network traffic is reserved for validation. All observations are labelled and contain a total of 200 observations of up to 58 attacks of different typologies, including different versions of denial of service (DoS), port scanning, and user-to-root (U2R) or remote-to-local attacks (R2L) [26].

These datasets, despite the year they were built, are still used today in various scenarios and their usefulness seems to be proven [36], although there are some studies that question their reliability [37].

#### 2.1.2. KDD Dataset

KDD99 [27] is a dataset created for the Third International Knowledge Discovery and Data Mining Tools Competition based on the DARPA dataset. Unlike the latter, KDD99 is a dataset whose format is based on the extraction of features (up to 41 [38]) from network flows rather than the recording of raw observed data. It is a synthetic dataset but takes into account the actual traffic observed in military network environments. Access to the dataset is open, and, despite its longevity, it is still available. In terms of size, the dataset contains almost 5 million observations, including the same typology of attacks as DARPA, i.e., DoS, port scanning and privilege escalation attacks.

Similar to DARPA, although it is a widely employed dataset, criticisms have emerged regarding its usability. Specifically, concerns have been raised about the lack of consistency between the number of attack types in the training subset and those available in the validation subset [39]. Additionally, the dataset is deemed outdated in the context of contemporary world communications.

#### 2.1.3. NSL-KDD Dataset

In 2009, to reduce the original DARPA and KDD problems, Tavallaee et al. [28] created a new version of KDD called NSL-KDD [28]. In this version, the authors removed all redundant records and added new synthetic ones based on the correctly labelled records of the original dataset, so that those record types with a lower presence in the original dataset had a higher presence in the new dataset and vice versa. As for the test dataset, it was completely regenerated. The result is a public dataset that is slightly more balanced, but with a very significant reduction in size, with just over 125 K observations in the training and 22.5 K in the testing set.

Even with the revision of the KDD dataset and the application of techniques to rebalance and address consistency issues, it continues to share the problems of its KDD and DARPA predecessors. Specifically, it relies on 1998 network traffic, rendering it outdated in the context of modern network communications and contemporary cyber-attacks.

#### 2.1.4. Kyoto 2006+ Dataset

Given the shortcomings of datasets such as DARPA and KDD with their variants related to the longevity of their data, in 2006, Song et al. [40] published a new dataset called Kyoto 2006+, the result of recording real traffic from 32 honeypots with different characteristics from November 2006 to August 2009 (almost three years), totalling more than 93 million observations [40]. Since its initial publication, the authors have expanded the dataset to cover a total of nine years of traffic (up to 2015), adding more honeypots to reach the final figure of 348, including DNS servers to generate benign traffic. Each record in the dataset provides a total of 24 features associated with the captured network traffic flows, of which a total of 14 are present in datasets such as DARPA or KDD, while the remaining 10 are new additions, including the labelling of the records, as well as the typology of the detected attack. This dataset is probably the public dataset of real traffic with the greatest historical depth on record, but, in spite of this, it is still quite balanced.

#### 2.1.5. Botnet Dataset

Biglar Beigi et al. [29] have developed a public dataset focused on botnet attacks, as they believed that this type of attack is currently the most challenging [29]. This dataset contains a total of 16 different botnet attack typologies, covering both centralised and decentralised attack strategies. In order to construct this synthetic dataset, the authors analysed different datasets by combining subsets of three different datasets (ISOT [41], ISCX 2012 IDS dataset [42], and *Botnet Traffic Generated by the Malware Capture Facility Project* or CTU-13 [43]) using the overlay methodology described in [44] that ensures the cohesion of the resulting data. The result is a dataset of tagged network packets with a total of almost 14 GB of information and a balance between normal and anomalous traffic of almost 55% and 45%, respectively, which is quite balanced.

#### 2.1.6. UNSW-NB15

The Cyber Range Lab at the Australian Centre for Cyber Security generated the synthetic UNSW-NB15 [30] dataset in 2015 using the IXIA Perfect Storm traffic generator. The simulation environment used to generate the samples consists of three servers, two of which generate benign traffic, while the third is used to generate traffic associated with various attacks such as DoS, exploits, and rootkits. The dataset size is reduced, reaching a total of 31 h in two subsets of 16 and 15 h, respectively, with just under 2.5 million observations, 12% of which correspond to anomalies or attacks. Labels are available for each flow, indicating whether it is normal or not, as well as the attack category to which it belongs. Finally, the data are available in packet format (PCAP) as a version of 49 features extracted from the captured flows.

#### 2.1.7. UGR’16

The UGR’16 dataset [31] was created by the University of Granada in 2016 as a result of capturing the real network traffic of a medium-sized ISP between March and June 2016. Subsequently, during the months of July and August, different attacks such as DoS, botnet, or port scanning were deliberately generated on the same ISP to capture all the traffic so that this subset could be used as a test. The dataset consists of NetFlow traffic flows with almost 17 billion different connections, of which more than 98% were normal traffic, making it very imbalanced. After the traffic was captured, state-of-the-art anomaly detection and network attack identification techniques were employed to tag the dataset. This involved assigning each record a label indicating the type of attack to which it belonged. Given the size of the dataset and its temporal proximity, it is an updated and current dataset for use in building or training AI and NIDS models.

As will be seen in more detail in Section 3.3, this dataset presents labelling problems, making it a good candidate to benefit from the methodology proposed in this paper.

#### 2.1.8. CIC Datasets

The Canadian Institute for Cybersecurity (CIC) has generated several datasets to validate the performance of NIDS or to train the models underlying these NIDS. Among the various datasets available, the following should be highlighted:CICIDS2017 [32]: Generated in 2017, it is a synthetic network traffic dataset generated in a controlled environment for a total of 5 days, available on request (it is protected). The captured data are in packet and flow formats, although they are also available in extracted feature format with a total of 80 different features. The captured traffic is tagged, and the different attacks that each record corresponds to, including DoS, SSH, and botnet attacks, are marked in the tag.CSE-CIC-IDS2018 [33]: This is a synthetic dataset generated in 2018 specifically based on network traffic intrusion criteria. It includes DoS attacks, web attacks, and network infiltration, among others, recorded on more than 400 different hosts. As with CICIDS2017, the data are in packet and flow formatw but with a version containing 80 extracted features, and access requires a prior request (protected). Unlike CICIDS2017, it is modifiable and extensible.

#### 2.1.9. NF-UQ-NIDS

Sarhan et al. [34] have created a synthetic dataset specifically created for machine learning-based NIDSs [34]. This dataset is the result of combining four datasets used in the NIDS domain but transformed into a netflow version. Two of the datasets used have been analysed previously in this work (UNSW-NB15 [30] and CSE-CIC-IDS2018 [33]), while the other two (BoT-IoT [45] and ToN-IoT [46]) are datasets generated by the Cyber Range Lab of the Australian Centre for Cyber Security (ACCS). The result is a dataset that contains flows from different networks with different configurations, making it more universal than the datasets of which it is composed. The original dataset to which each flow belongs is available, allowing us to know under which scenario or network a NIDS trained with NF-UQ-NIDS can be more or less effective. The dataset contains almost 12 M records, 76.77% of which correspond to normal traffic, while the remaining 23.33% correspond to the 20 types of attacks it contains, making it an imbalanced dataset. It was published in 2021, so it is a dataset that can be considered up-to-date and incorporates the latest types of attacks.

### 2.2. Dealing with Labelling Problems in Datasets and the Techniques to Address Them

Classification problems, whether supervised or unsupervised learning, require a sufficiently large dataset that is correctly labelled. In the case of supervised learning, the labelling is used so that the model learns to distinguish the different classes that make up the universe being treated. However, when the problem is approached from an unsupervised learning perspective, such as anomaly detection, the training dataset is expected to belong to the same class. This setup enables the model to learn to identify anomalies by recognizing deviations from the patterns present in the training set. The process of creating a dataset is therefore very important, as it determines the potential success of the machine learning models that will use it.

The processes of tagging the data that make up a dataset involve the application of automated techniques, as well as manual processes, which together can be subject to error [47]. To mitigate this problem, some papers present methods or techniques to reduce the mislabelling that occurs. For example, Kremer et al. [48] propose a model that tries to detect the noise in the labelling based on loss functions that are insensitive to noise and at the same time tries to infer the possible noise in the labelling and in the classification itself [48]. On the other hand, Zhang et al. [49] propose a framework called Adaptive Voting Noise Correction (AVNC), which aims to identify and correct incorrect labelling [49]. However, even the application of these techniques does not guarantee the correct labelling of the dataset.

When the labelling of the data that make up a dataset is performed manually, there is a risk of unintentional bias that is intrinsic to the observed data. To address this scenario, a methodology is proposed in [50], whose aim is to relabel the data, eliminating the possible bias of the initial labelling, achieving good results in a computational perception problem on galaxy detection.

The impact of noise on labelling in artificial intelligence models has also been analysed in several works in a way that relativises its impact. For example, Natarajan et al. [51] propose in [51] a simple loss estimator that is unbiased and minimises the risk of the presence of mislabelled data. Another approach, as proposed by Patrini et al. [52], focuses on tackling the issue of noise in labelling, particularly in scenarios involving deep learning models, including recurrent neural networks. The authors suggest two procedures to correct the loss function in instances of mislabelled data [52]. More recent is the work of Wei et al. [53], who this problem and propose two datasets with noise in the labelling to serve as a benchmark to measure how robust the models or techniques are to errors in the labelling [53].

Of particular relevance is the work of Northcutt et al. [47], which analyses the quality of labelling in test subsets of 10 datasets, as opposed to the work presented above, which focuses on the quality of labelling of the training data. This approach is particularly interesting as the test subsets are assumed to be perfectly labelled, as they are the test and evaluation mechanism by which the models are tested and validated [47]. Labelling errors in such a dataset can destabilise the performance of machine learning models. The datasets tested are those commonly used in the field of computational perception (such as MNIST or ImageNet), in the field of language processing (such as IMDB or Amazon Reviews), and finally in the field of audio processing (AudioSet). The results obtained show that there are labelling errors that, in some cases, reach up to 10% of the labelling error.

Confident Learning (CL) is a subfield of machine learning between supervised and semi-supervised learning that focuses on characterising noise in the labelling to find and correct errors in the labelling in order to train robust models. To achieve this, they use data-pruning techniques to clean the dataset before training the models. In [54], a generalised CL strategy is proposed that is able to find the errors in the labelling by estimating the correct distribution of correct and incorrect labels. Furthermore, it is tested on image datasets, yielding models with higher performance than some of the best state-of-the-art models.

Müller and Markert [55] propose a tool to detect errors in the labelling of image, text and numerical datasets [55]. As a result of the application of this tool, the set of observations of the dataset with a high probability of being mislabelled is obtained. This method has been tested on a total of 29 different datasets, both real and synthetic and, according to its authors, has been able to find mislabelling in some of them that had not been detected before.

The application of computational perception techniques in medicine is also subject to the risks associated with mislabelling, especially when the goal is to detect the presence of possible tumours. In [56], the authors addressed this problem by proposing a methodology to identify labelling errors in images associated with the presence of breast cancer. To achieve this, they propose a function that measures the deviation between the prediction made by the model and the real value of the sample (called Cross-Entropy loss). Additionally, they put forward another function that assesses the model’s dependence on the dataset, known as the Influence function. The method is evaluated on a set of 10,500 images in which up to 98% of labelling errors are detected.

Another methodology in the field of image processing is proposed in [57], where the aim is to train a deep learning model with a dataset where there is no confidence in the labelling of the data. To do this, the model adjusts the internal parameters of the neural network while learning the distribution of noise in the labelling and testing it against classical back-propagation models where the goodness of the labelling is assumed.

In the specific area of datasets aimed at addressing cybersecurity or network traffic problems, previous work is more limited, as the generation of these datasets has additional complications with respect to the more general use cases. In [58], Cordero et al. [58] the problem is reviewed through a comprehensive analysis of various datasets intended for NIDS. The authors put forth an enhancement to the Intrusion-Detection Dataset Toolkit (ID2T) dataset generation methodology. Subsequently, they evaluate the effectiveness of the proposed ID2T improvement by assessing datasets generated after its application.

The problem of labelling in the field of network traffic is more complex, since it requires specific low-level knowledge of the traffic in order to be able to correctly classify each flow. In [59], an analysis of the methods used for labelling this type of dataset, both automatic and manual, is carried out, identifying the weaknesses of each of the techniques along with their advantages and disadvantages.

Finally, to conclude this analysis of the state of the art in dataset quality, in [60], an approach to measuring the quality of a network traffic dataset is presented. This quality is used to compare two datasets, to decide if they are equivalent, or if a better quality dataset is found, whether or not it is appropriate to retrain the machine learning models. The proposal for measuring the quality of a dataset is based on the criteria: (i) completeness as the probability that a dataset record can occur in the domain of the machine learning model to be built and (ii) reliability as the probability of occurrence of misclassified or mislabelled data for each possible class. Based on these two criteria, the applicability of a network traffic dataset to a particular problem can be determined.

## 3. Materials and Methods

### 3.1. Kitsune NIDS

Kitsune [9] is a particular case of autoencoder applied to network intrusion detection. According to its authors, it is a NIDS designed to efficiently detect anomalous network traffic, computationally simple, and feasible to be deployed on any router for real-time detection. These features allow Kitsune to be used in experimental scenarios where the number of simulations is high, reducing execution times and computational cost.

The architecture of Kitsune, shown in Figure 1, is composed of five entities or components:**Packet Capturer**: This is not an integral part of the Kitsune solution but rather a third-party library or application that captures network packets in pcap format.**Packet Parser**: Like the capturer, this module also corresponds to a third-party library (e.g., tshark or Packet++ [61]), and its function is to extract metadata from raw network packets, for instance, the packet’s source and destination IP addresses, timestamp, ports involved, and packet size.**Feature Extractor**: This is the first component that is part of Kitsune. Its purpose is to extract a set of *n* numeric features that accurately represent the channel status through which the packet was received and maintain a set of statistical data representing the traffic patterns collected so far, thus linking the time sequence of traffic to the detection process.**Feature Mapper**: In this component, dimensionality reduction is executed to transfer each observation from a set of *n* features to a smaller and more concise set of *m* features while preserving the correlation between *n* and *m*. To accomplish this, segmentation or clustering techniques are adopted to categorize the features into *k* groups, each with no more than *m* traits. These groups will be employed in each autoencoder that makes up the anomaly detector.**Anomaly Detector**: The final step is to assess whether each observation constitutes an anomalous packet or not.

The Anomaly Detector, referred to by its authors as KitNET [62] (***Kit**sune**NET**work*), can be used independently as an anomaly detector if data are available in the form of continuous feature vectors.

KitNET is an unsupervised learning neural network designed for real-time anomaly detection. To carry out this task, its architecture is divided into two clearly differentiated elements:**Ensemble of Autoencoders**: This is a set of *k* identical autoencoders, formed by three layers whose input has *m* neurons corresponding to the *m* characteristics chosen by the feature mapper. The objective of this ensemble is to measure the degree of anomaly that each observation has independently. To do this, the root-mean-squared error or RMSE (eq:rmse) of the observation is calculated at the output of each autoencoder. All observations go through all autoencoders, although only a subset of *m* characteristics of the observation is used in each one of them.
(1)$$RMSE=\sqrt{\frac{\sum_{i=1}^{n}{\left(predicted_{i}-actual_{i}\right)}^{2}}{n}}$$**Output Layer**: This is also a three-layer autoencoder that receives as input the output of the entire previous ensemble (the RMSE generated by each autoencoder) and whose output is also an RMSE that is transformed into a probability by applying a logarithmic distribution. This probability indicates the degree to which the observation is considered to be anomalous traffic.

From the perspective of KitNET operation, the execution sequence is divided into four phases:**Initialisation**: Based on a set of observations, the model calculates the number of autoencoders that will form the ensemble and the features that will be used in each of them.**Training**: Using a new set of observations, the internal parameters of each autoencoder are adjusted to reduce the RMSE generated in their output layers. This phase is subdivided into the following:(a)**Calibration**: The weights and internal parameters of the neural networks of the autoencoders are adjusted using a subset of the provided training data.(b)**Testing**: After the previous process, the remaining data in the training set are used to generate a probability distribution from which to choose the threshold that will distinguish a normal observation from an anomalous one.**Detection**: Once the internal parameters of the autoencoders are fixed, each new observation that passes through the architecture generates an RMSE in each autoencoder of the ensemble. All these values serve as input to the output layer of the model.**Labeling**: The final result that KitNET produces is the calculation of the probability that the observation is anomalous. Using the threshold obtained during training, it is determined whether that probability makes the considered observation anomalous or not.

Finally, it is important to highlight that KitNET (and therefore Kitsune) assumes that the data used for the initialisation and training phases are observations of normal traffic, that is, free of anomalies.

### 3.2. Feature as a Counter

Captured network traffic generally consists of a large number of files in binary format with diverse information that cannot be directly used in machine learning algorithms [63]. Therefore, an aggregation process is necessary to obtain a universe of data that can be handled as a dataset.

Camacho et al. propose in [64] a technique called *Feature as a Counter* (FaaC). The primary objective of FaaC is to convert a dataset related to network traffic, specifically network flows, into a singular observation matrix. This matrix is designed to be suitable for multivariate analysis or as input for machine learning algorithms in a customized manner. With this tool, captured network traffic flows from different sources can be flexibly combined and aggregated such that new observations are formed from new counting variables of certain values present in the original variables. For example, it may be interesting to know the total number of connections per minute directed to ports 80 and 443 of a web server, so that if a very high number is recorded in a certain time window (one minute by default), it could be a symptom of a *Denial of Service* (DoS) attack. To do this, FaaC allows converting the raw data of all captured network flows (timestamp, source and destination IP address, source and destination ports, packet size, flags, etc.) into a dataset in which, for each minute, the number of flows using port 80, 443, 22, etc., is counted.

A specific implementation of FaaC can be found in a library called FCParser [65], which has been applied to calculate minute-aggregated counters for all available observations in UGR’16. In this aggregation, a total of 134 new counting variables or features are determined for each minute, including the different most common ports of both origin and destination (*FTP*, *SSH*, *SMTP*, *HTTP*, etc.), the different protocols involved (*TCP*, *UDP*, *ICMP*, etc.), the different flags of *TCP* packets (*ACK*, *RST*, etc.), the priority and size of the packets, and the total flows of each type of anomaly.

### 3.3. UGR’16 Dataset

UGR’16 is a dataset presented and analysed by Maciá-Fernández et al. in [31]. It is divided into the following subsets:**Training set** (calibration): Real traffic data observed in an Internet Service Provider (ISP) during the four months from March to June 2016.**Test set**: Real traffic data observed in the same ISP and synthetically generated attack traffic during the two months of July and August 2016.

Since the data source is the traffic collected from the ISP’s facilities, the heterogeneity of the captured records is high. These records encompass data associated with a wide variety of communication protocols (HTTP, FTP, DNS, etc.) and originate from various user types. Importantly, the data are not confined to a specific set of actors belonging to a single company, university, or laboratory, contributing to the overall diversity of the dataset.

For the synthetic attack data, simulated Denial of Service (DoS) attacks, port scanning, and attacks caused by malware (specifically, botnet attacks) are included. These attacks are concentrated to run on specific days between July and August, using two strategies: (*i*) fixed execution dates and times, and (*ii*) random attack time (see [31] for more information).

As an additional feature of UGR’16, anomalies were detected in the dataset related to SPAM campaigns, SSH port scanning, and UDP port scanning. Both synthetically generated attacks and detected anomalies were properly labelled in the dataset using state-of-the-art techniques. Additionally, traffic belonging to IP addresses on blacklists was also labelled.

An inherent problem with the capture of real traffic is that it is possible that real attack traffic flows may already be present in the training data but have gone unnoticed and have been erroneously labelled as normal or background traffic. This can introduce noise into the dataset and degradate its quality, as it may be computing traffic flows as normal that actually are not, thereby hindering the detection methods being built. In fact, García Fuentes, M., in his doctoral thesis “Multivariate Statistical Network Monitoring for Network Security based on Principal Component Analysis” [19], identifies in the UGR’16 dataset, specifically in the calibration part of June, an undetected botnet related attack. This makes anomalous traffic appear as normal traffic.

In [20], Medina-Arco et al. confirmed the existence of a labelling error in UGR’16 in June using KitNET. They further identified a labeling problem due to the botnet attack that started in the last days of May, revealing hidden anomalies that had not previously been identified. These findings stem from investigations into KitNET and UGR’16. Various scenarios were set up to investigate the incremental use of calibration dataset data. These were (*i*) training with March data and validation against July and August data; (*ii*) training with March and April data compared to July and August test data; and (*iii*) training with March, April, and May data compared to July and August test data.

Table 2 presents the traditional performance metrics achieved by applying KitNET in all scenarios. It is worth noting a decline in the values of the third scenario as the month of May was included in the training. It is particularly noteworthy that the recall and F1-measure values both drop. Recall drops to 23%, while F1-measure drops to 36%. Table 3 displays the detection rates by attack type. It is clear that the inclusion of May results in a decrease in the detection capacities for all attack types. This decrease is particularly significant in the case of botnet attacks, where only 1% of them couold be detected compared to the 74% detected in the previous scenario.

Figure 2 shows the ROC curves for the three scenarios. Notably, the scenario including May in the training exhibits inferior behaviour compared to the previous scenarios, with an AUC (area under the curve) that is much lower (0.89) compared to the other two (both around 0.95).

Finally, the analysis of network traffic linked to botnet-style attacks associated with the IRC protocol (presented in Figure 3) indicates a considerable rise from the final days of May to the middle of June (dashed green square in the figure). This increase suggests that the attack, which was not detected by the UGR’16 labelling process, probably took place during this time.

## 4. Proposed Methodology

In this section, we outline the main procedures of the proposed methodology aimed at identifying hidden anomalies or tainted data in the training data used in a machine learning model in the field of NIDS. The Figure 4 represents the proposed methodology, which will be detailed below.

First, it is necessary to estimate the optimal number of observations in the entire training dataset that is ideal for training an AI model. It is then necessary to identify the subset of training data of that size which maximise the model’s performance while enabling the potential discovery of hidden anomalies within the dataset. To achieve this, metrics are established to measure the quality of each data subset. The stopping mechanisms for the proposed process will then be determined based on these metrics.

Searching for the optimal window size is separated from the search for the optimal window to minimize the total number of scenarios to be evaluated. This approach makes the overall process more efficient while preserving the temporal sequence of the data recorded in the dataset. In the second step, the entire training dataset is analyzed to search for the optimal subset of training data which allow the discovery of potentially hidden anomalies. Considering *n* as the number of blocks into which the dataset is divided (hours, days, weeks, etc.), the maximum number of scenarios needed for evaluation with this strategy is 2n. Therefore, the computational complexity is linear (O(n)).

At the end of this section, we also propose an alternative method for performing an exhaustive search for the optimal window in the training dataset.

### 4.1. Performance Metrics

Among the most frequent measures for assessing the efficacy of a binary classification model, like an anomaly detector, are *precision*, *recall*, *accuracy*, and *F1-score*. These metrics are obtained by deriving the observable indicators from a classifier’s predictions on test records and counting the number of true and false positives and negatives (True Positive (TP), True Negative (TN), False Positive (FP), False Negative (FN)).

Among these indicators, the most relevant one in cases where the dataset is highly imbalanced, as is typically the case for datasets utilised in the NIDS domain, is the *F1-score* [66]. This indicator is a combined measurement of *precision* and *recall*, which is defined by Equation (Equation 2). *Precision* quantifies the proportion of positive predictions that are correct, whereas *recall* measures the fraction of positive observations that the model has detected with respect to the total positive observations in the dataset. In datasets wherein the number of positive samples is considerably smaller in comparison to the total number of negative samples, these metrics may not adequately measure the quality of a model. For instance, imagine a dataset with 90 samples of *class A* and 10 samples of *class B*. If a model solely predicts elements of *class A*, it will have an accuracy of 90%. However, its performance would be incorrect since it cannot detect any instances of *class B*. Meanwhile, *recall* would achieve a result of 100%, yet this does not demonstrate the model’s effectiveness. *F1-score* aims to address this issue by presenting a harmonic mean between *precision* and *recall*. This means that the higher the *F1-score*, the higher the *precision* and *recall* are, jointly decreasing the impacts resulting from an imbalanced dataset.
(2)$$f1=\frac{2TP}{2TP+FP+FN}$$

Another useful metric for selecting the best model or algorithm for classification problems is the AUC value (Area Under the ROC Curve), which compares the true positive and false positive ratios. The ROC curve (Receiver Operating Characteristics) associated with this value determines the overall probability of classifier success for both classes [67], as shown before in Figure 2. The AUC value positively correlates with the likelihood of accurate predictions made by the model in any of the problem’s classes, namely normal traffic or anomalous traffic.

Both the F1-score and AUC are utilised in the proposed methodology to evaluate the efficacy of the diverse models produced at every phase of the procedure.

### 4.2. Optimal Number of Observations for Training

As discussed in Section 2.1, actual network traffic datasets are generally vast, capturing a wide variety of traffic in order to identify all patterns associated with normal and anomalous usage in a given environment. The datasets are intended to capture the maximum amount of traffic of all possible types. The result is a training set that could potentially be remarkably large, but as demonstrated in the specific case of UGR’16 (see Section 3.3), there is no guarantee that a greater number of observations in a training set will produce superior results in an artificial-intelligence-based model. Therefore, it seems reasonable to ask what the optimal number of training data is, in terms of the number of observations, for a particular dataset in order to maximise the performance of a model trained on that dataset. Moreover, refining the number of records utilized for training leads to enhanced efficiency in the training process. This results in computational ease and faster execution of the process.

In the realm of network traffic, unlike other artificial intelligence problems where it may be feasible, and even recommended, to implement random data selection techniques within the training set for the purpose of training a machine learning model, the traffic sequence plays a critical role [68]. Indeed, it is this temporal sequence that records the patterns of behaviour and network usage. Changing the order of events could affect model performance to varying degrees, but it would fundamentally alter the essence of the actual traffic that exists in genuine traffic datasets. This, in turn, leads to the emergence of patterns that could not exist in a production environment due to their lack of logical coherence. Randomly selecting network packets from a genuine traffic dataset would, for example, disrupt the established sequence of a TCP conversation flow.

Due to the inherent specificity of network traffic and the goal of acquiring an optimal set of records to train an anomaly-detection model, the first step of the proposed method is to identify the optimal size of a training data sequence, referred to as the *training window*.

#### 4.2.1. Finding the Best Size for the Training Window

To determine the ideal training window size, the proposed method utilises a process of iterative training. In each iteration, the number of records used increases progressively, starting from the first record of the dataset subset designated for training. At the end of training in each iteration, the resulting model is tested on the designated test dataset, with an evaluation of the *F1-score* and *AUC* indicators.

Figure 5 presents an example of how this process can be applied. It is important to note that each identified unit is equivalent to, for instance, a recorded day of data in the dataset. Therefore, during the first iteration, the model will be trained solely using data collected on the first day of the training dataset. The model is validated using the full set of test data available. From these results, we will derive the *F1-score* and *AUC* values. In the second iteration, the model will be trained using the initial two days of training data and subsequently retested with the complete test dataset to obtain the *F1-score* and *AUC* values again. We then repeat this process iteratively until training has been carried out on the entire set of training data.

The optimal size of training window will be determined by the iteration that yields the highest *F1-score* and *AUC* values.

#### 4.2.2. Early Stopping

Early stopping conditions can be implemented in this iterative and incremental training process to decrease the time required to search for the optimal window size for training. To achieve this, it is enough to set improvement thresholds for certain indicators used to measure the quality of each iteration (such as *F1-score* and/or *AUC*), similar to how it is measured in training machine learning models. If these indicators do not improve beyond the defined threshold within a specific number of iterations, it indicates that the model might be at a local minimum of the gradient descent [69]. If, after a certain number of iterations, the values of the *F1-score* and *AUC* do not surpass the maximum values obtained so far, the algorithm will stop, and the optimal scenario will be identified as the one that achieved those maximum values. This avoids further iterations until the entire training set has been used.

The sole risk related to utilizing early stopping criteria is that the process may halt at a local minimum, rather than the global minimum. As a result, it might overlook a more significant window size that yields superior results. Nevertheless, the second step of the proposed process remedies this problem by sliding the window, thereby covering the entire training dataset.

A potential approach to implement this step utilizing early stopping is demonstrated in Algorithm 1.
**Algorithm** **1** Step 1: Finding the optimal number of observations for training**Require:** Training dataset, Test dataset and incremental unit (number or flows/pactkets,   hours, days, weeks, …)
**Ensure:** The (pseudo) optimal window size  max_f1_score,max_auc,optimal_windows_size←0  unimproved_scenarios←0  early_stopping←10  stop←false  current_size←incremental_unit  **while** stop≠true **do**     model_train(training_data(begin,current_size))     f1,auc←model_predict(test_data)     **if** f1≥max_f1_score or auc≥max_auc **then**         max_auc←auc         max_f1_score←f1         optimal_windows_size←current_size         current_size←current_size+incremental_unit         unimproved_scenarios←0     **else**         unimproved_scenarios←unimproved_scenarios+1     **end if**     **if** unimproved_scenarios≥early_stopping or current_size≥last_dataset_date **then**         stop←true     **end if**  **end while**

### 4.3. Optimal Data Window Size for Training: A Sliding Window Approach

After completing the previous step, a training window is obtained. This window yields optimal results when applied from the beginning of the training dataset. Nevertheless, it cannot be guaranteed that it is the best window to maximize the performance of a machine learning model of a NIDS that employs this dataset for training. It is important to consider that the aim of this study is to identify hidden anomalies in a genuine traffic dataset. Therefore, the outcome from the previous phase may not adequately fulfil this objective. If unreported or unlabeled anomalies are present in the initial training entries, what impact would it have on the analysis? The findings of the previous step suggest that increasing the window size may dilute the impact of labelling errors caused by a larger training sample size. However, it would still not eliminate the contamination and insensitivity of the model towards attacks caused by hidden or mislabelled anomalies.

To address this issue, the proposed methodology presents a secondary step that intends to achieve two objectives:Identify the ideal training data window or sequence from the training dataset to maximise model performance. This dataset will be used to train any machine learning model-based NIDS that uses it.Analyse the performance of different models trained on each subset of the training set to uncover hidden anomalies in the dataset.

This second stage of the proposed methodology involves sliding the training data window across the entire dataset to assess the goodness of fit of the training dataset.

#### 4.3.1. Finding the Optimal Data Training Window

After determining the optimal training data window size based on the preceding step, the proposed approach utilises a repeated training process, whereby the data window used for training is shifted through the entire training dataset during each iteration. At the conclusion of each cycle, the resulting model is evaluated using the test dataset, with performance indicators including *F1-score* and *AUC*. The ideal training data timeframe is the one that delivers the maximum *F1-score* and *AUC* values.

Figure 6 graphically illustrates the proposed process. Given that the optimal training window is, for instance, of five units (e.g., days), the following steps should be followed:**Step 1**: The model is first trained using the training data from the initial 5 days of the dataset. The model is then re-validated against the test data to generate the *F1-score* and *AUC* values.**Step 2**: In the second step, the model is trained with the training data obtained from the subsequent 5 days of the dataset (day 1 to day 6), by sliding the window by one day. Afterwards, it is validated against the complete test dataset, and the resulting *F1-score* and *AUC* performance indicators are obtained.**Step n**: These steps are repeated for all subsequent iterations (*n*). The training window is shifted successively until it occupies the final 5 days of the training dataset. The *F1-score* and *AUC* are calculated as in other iterations.

After analysing the results of all iterations, the subset of data used for training that performed best is considered optimal for training an anomaly-based NIDS.

#### 4.3.2. Hidden Anomaly Detection

As an additional result to the search for the optimal window of training data with which to train the model of a NIDS, a tool is obtained that allows analysing the existence of possible errors in the labelling or the presence of unidentified anomalies in the dataset.

Under normal conditions, where the labeling of a network traffic dataset is correct, an artificial intelligence model is expected to exhibit equivalent results when trained with subsets of data that are similar in size, even if they comprise different days of that set. This is because the model’s strength lies in its capacity to generalize the data, enabling it to infer the knowledge necessary for detecting whether a future record is normal or not. This would imply that the *F1-score* and *AUC* results for each of these possible subsets should be, to some degree, similar.

However, when unlabelled data are present and a model is trained on these data, it is likely that the *F1-score* and *AUC* results obtained when confronted with the test data will be drastically worse, as it will not be able to detect some of the anomalies. This effect is produced by the fact that the model has inferred that a certain correlation of data and patterns is considered normal when in fact it is not.

Therefore, the analysis of the evolution of the *F1-score* and *AUC* metrics can help to detect possible sections of the dataset that contain potential labelling problems and, therefore, facilitate the task of analysis by subject matter specialists to determine the source of potential quality problems in the dataset.

A potential approach to implement the Step 2 can be found in Algorithm 2.
**Algorithm** **2** Step 2: Finding the optimal data window for training**Require:** Training dataset, Test dataset and window size (number or flows/pactkets, hours,   days, weeks, …)**Ensure:** The optimal training window  max_f1_score,max_auc←0  start_window,end_window←0  optimal_start_window,optimal_end_window←0  iteration←0  **while** end_window≠end_training **do**     start_window←start_training+iteration     end_window←start_training+window_size+iteration     model_train(training_data(start_window,end_window))     f1,auc←model_predict(test_data)     **if** f1≥max_f1_score or auc≥max_auc **then**         max_auc←auc         max_f1_score←f1         optimal_start_window←start_window         optimal_end_window←end_window     **end if**     iteration←iteration+1 **end while**

### 4.4. Exhaustive Search for Optimal Window

The use of early stopping techniques, as described in Section 4.2.2, can locate a window that corresponds to a local minimum and is, therefore, a sub-optimal solution. To ensure that the optimal solution is located, it is necessary to perform an exhaustive search that combines the start of the window at every possible location in the dataset with all possible window sizes.

Figure 7 shows an example of how this exhaustive search could be performed. Consider, for example, *n* as the total number of days contained in a dataset. In the first iteration, the starting point of the window is considered to be the first day of the dataset and has a depth of only one day. This window is used to train the model and validate it against the test dataset. As a result, the *F1-score* and *AUC* metrics are obtained. In the next iteration, the start of the window is maintained and the depth of the window is increased in one day to repeat the exercise. The process is repeated until the end of the window is reached, which corresponds to the last day of the dataset.

The next step is to shift the start of the window by one day and repeat all iterations until the entire remaining dataset is used. This process will be repeated until the start and end of the window coincides with the last available day of the dataset.

The combination of window start and window depth with the best F1-score and *AUC* results represents the optimal training data subset that maximises model performance.

The total number of scenarios required to run the optimal window search is as defined in Equation (Equation 3), so its computational complexity is $O(n^{2})$. It is therefore computationally worse than the two-step search strategy, which was linear.
(3)$$\frac{1}{2}n\left(n+1\right)$$

A potential approach to implement this exhaustive search is demonstrated in Algorithm 3.
**Algorithm** **3** Exhaustive search for the optimal data training window**Require:** Training dataset, Test dataset and window size (number or flows/pactkets, hours,   days, weeks, …)**Ensure:** The optimal training window  max_f1_score,max_auc←0  start_window,end_window←0  optimal_start_window,optimal_end_window←0  iteration←0    **for** i←1 to *n* **do**        **for** j←i to *n* **do**        start_window←start_training+i        end_window←start_training+j        model_train(training_data(start_window,end_window))        f1,auc←model_predict(test_data)        **if** f1≥max_f1_score or auc≥max_auc **then**            max_auc←auc            max_f1_score←f1            optimal_start_window←start_window            optimal_end_window←end_window        **end if**    **end for****end for**

## 5. Results

The current section tests the previously proposed methodology in a scenario that combines a dataset with the presence of hidden anomalies, such as UGR’16, and the state-of-the-art NIDS Kitsune. To accomplish this, the experimental scenario is first defined, followed by a demonstration of the obtained results.

### 5.1. Description of the Validation Scenario

The application of the proposed methodology in a real use case to validate its applicability and to be able to analyse the results obtained has been carried out with the following considerations:Instead of using UGR’16 packets, the data are represented by numerical features derived from the Feature as a Counter method, as explained in Section 3.2.Out of the entire UGR’16 dataset that was allocated for training, any observations corresponding to attacks were removed. This was performed in order to create a training dataset that is free of anomalies, which is a requirement for KitNET.The state-of-the-art NIDS applied in this experimentation scenario is Kitsune (Section 3.1), although its application is reduced to the use of the Anomaly Detector (KitNET) together with the features extracted from UGR’16.The specific configuration parameters for the KitNET model are as follows:–The maximum size of each autoencoder in the internal ensemble is empirically set to 10 neurons in the hidden layer.–The number of instances from each scenario’s training dataset used for the initialisation phase of KitNET is empirically set to 2000.–The proportion of instances from every training dataset scenario used for KitNET’s training sub-phase is set to 70%.–The proportion of instances used for the KitNET validation sub-phase, therefore, is 30%.–A uniform value for the threshold or tolerance threshold for anomaly detection has been utilized in all cases—the standard deviation added to the mean of the probabilities detected during the training phase. Values above this threshold are considered anomalous.Since we have defined for KitNET 2000 observations for the initialisation phase, the initial window must be at least 2 days long.In each iteration, the window for the first step of the methodology is 1 day.Instead of using the exhaustive search in order to minimise the computational cost, an early stopping mechanism is implemented based on a total of 10 iterations, which improves neither the *F1-score* nor the maximum *AUC* found so far.

### 5.2. Experiment Results

#### 5.2.1. Step 1—Looking for the Window Size

After 50 iterations, the search for the most appropriate training window size for UGR’16 was completed. The early stopping mechanism was ultimately activated when there was no further improvement in the results obtained so far.

The findings from the performance metrics of the KitNET model with UGR’16 for varying training window sizes are showcased in Figure 8. *Accuracy*, *Precision*, *Recall*, *F1-score*, and *AUC* values can be reviewed. The window size that maximizes the *F1-score* and *AUC* values is indicated by a blue dotted line, as well as the maximum values of *F1-score* and *AUC*.

#### 5.2.2. Step 2—Looking for the Optimal Training Window

After determining the optimal size of the window (40 days in this example), the second step of the methodology is initiated, which comprises 58 iterations. During each iteration, the model is trained using 40-day blocks of training data, starting from the next day in each iteration and sliding one day at a time until the last 40 days of data are used in the training dataset.

Figure 9 displays the results acquired for the *AUC* and *F1-score* metrics for every iteration, indicating the iteration with the highest *AUC* and *F1-score* values, and the maximum values achieved by them are delineated with a blue dotted line. The value on the x-axis indicates each iteration, which in turn represents the number of days the window has been shifted from the origin. It should be noted that when the iteration’s value is 0, the dataset used for training covers the first day of the data in the training dataset up to 40 days later (where 40 days is the optimal window size obtained in the previous step of the methodology).

## 6. Discussion

Upon completion of the proposed experimental case methodology, examination and discussion of the acquired results can take place. The goal is to confirm the efficacy of the procedure proposed in this study. However, as anticipated in Section 2.2, there are no other references in the literature aimed at finding the optimal subset of data for training while searching for hidden anomalies.

### 6.1. Results on Looking for the Training Window Size

Firstly, we address the initial step of the methodology, which seeks to find an ideal window size for training an anomaly detection model in a NIDS, as shown in Figure 8.

The accuracy values are very close to 100%, but this does not necessarily mean that the model is optimal overall. This is because the UGR’16 dataset is heavily imbalanced, with only 2% of the observations corresponding to attacks. As a result, high accuracy rates are only achieved by having a high hit rate in the normal traffic class. However, the main objective of a NIDS is to detect the anomalous traffic representing 2%. Therefore, this indicator cannot be considered as a measure of model quality.

In the initial iterations, there is significant oscillation in the remaining performance metrics. This phenomenon could stem from inadequate training, namely the insufficiency of the available data to enable the model to deduce all probable cases. Consequently, its aptitude to generalize correlations between data, to make predictions in the future, is still in its early stages. This is supported by iteration 8 onwards, where the model received over 8 days of normal traffic. This resulted in the stabilization of performance metrics.

After 20 days of training, the model showed a qualitative leap in all performance indicators. This suggests that generalization is consolidated and the model could correctly classify 80% of the observations, as indicated by the AUC.

At this point, the model has shown marginal improvement in performance, suggesting that it may have achieved the maximum level of generalisation for this data set, having peaked at iteration 40.

Just after this point, when the training window reaches 3 May 2016, the performance significantly drops following the same pattern in all performance metrics. The effectiveness decreases by more than 15%, indicating that something anomalous is occurring on those days. The drop could result from a traffic pattern that, although normal, is unlikely to occur frequently or it might be due to an undetected attack, which is then to be considered as part of the normal traffic.

In the subsequent iterations, despite a minor rebound, the model does not exhibit any improvement from the earlier data and ultimately ceases the process by implementing the early stop criteria.

### 6.2. Results on Looking for the Optimal Training Window

Once the optimal window size has been determined, the next step is to establish the most fitting sequence of that size in the training dataset. Iterations are then performed throughout the training dataset.

In Figure 9, the evolution of the *F1-score* indicates that the optimal window covers the initial 40 days of the UGR’16 training dataset. This is due to a significant decrease in the indicator as the window is shifted within the dataset. However, the *AUC* remains relatively stable in the initial iterations. This could be explained by the imbalanced nature of the dataset.

In iteration 18, which covers the training data from 6 April 2016 to 19 May 2016, a 50% decrease in the *F1-score* is observed, resulting in a value of 0.32. This decrease suggests that the model is struggling to predict anomalous traffic and its ability to accurately predict normal traffic is decreasing. This is supported by the drop in *AUC* values during the same period. The cause of this behaviour may be the presence of observations or records in the training dataset, which create confusion for the model. As a result, the model is unable to accurately classify with quality. This indicates the potential for misclassified data to emerge starting on May 19, coinciding with the onset of botnet attack activity as depicted in Figure 3.

This pattern continues until the lowest level of performance is achieved globally on the twenty-ninth iteration. The training data utilized in this case cover the period from 16 April 2016, to 29 May 2016. The trained model is only able to achieve an *F1-score* value of 0.08 when evaluated with this dataset, rendering it almost entirely incapable of identifying anomalous traffic. It appears that the largest number of mislabelled records or undetected anomalies can be found during this time period, as the model struggles to identify anomalous traffic despite being trained on these data. As a result, there is a noticeable impact on the AUC, with its ratio decreasing to 0.8.

Subsequent iterations show a slight improvement over the global minimum detected. However, performance remains significantly low. Consequently, models trained with these subsets of data would not be appropriate for use in a production NIDS. These scenarios span the entire month of June and coincide with the appearance of an unidentified botnet attack, which could provide an explanation for the poor performance detected.

The most recent scenarios exhibit a noteworthy enhancement, although they are still incapable of recuperating from the initial performance drop. This is attributable to the window size employed, which encompasses the unlabelled June botnet attack observances as part of the training. Nevertheless, considering the trend displayed in Figure 9, it is probable that the system’s performance can recover to the peak levels achieved in the preliminary scenarios if the dataset contained more samples beyond June.

To further analyse of the results obtained in this phase of the methodology, Figure 10 shows the evolution of the total *F1-score* metric, as well as the *F1-score* metric solely for botnet-related attacks. Based on the given information, it seems that the decline in the *F1-score* of the model corresponds to the significant decrease in the *F1-score* related to botnet attacks. This implies that there are hints of botnet attacks present in the training set of UGR’16 by 19th May 2016 that remain undetected.

## 7. Conclusions

In this paper, we present a methodology for detecting contaminated data in actual source network traffic datasets for training NIDS-based anomaly detection. This methodology has three primary aims: (*i*) first, to determine the optimal size of the training dataset subset, which enables the NIDS model to perform best regarding anomaly detection; (*ii*) second, to choose the data subset from the training dataset that attains maximal performance for the machine learning model of the NIDS used for anomaly detection; (*iii*) third, to examine the dataset quality and identify potential labeling problems or polluted data by searching for the aforementioned training subset.

The UGR’16 contaminated dataset was tested with the NIDS Kitsune using this methodology, resulting in the identification of the ideal data subset to improve the efficiency of Kitsune. Additionally, potential botnet attacks were discovered in May that were previously undetected, and labeling errors in June were confirmed.

Future work will tackle the application of this methodology to some other real network traffic datasets and in combination with other state-of-the-art NIDSs. Altogether, it will contribute to probing the suitability of our proposal in searching for optimal subsets of data to robust anomaly detectors. Jointly optimizing both the window size and location in the dataset using well-known metaheuristics such as Particle Swarm Optimization (PSO), could be a viable approach for analysis. It might also be interesting to validate the applicability of this methodology in the face of poisoning-type adversary attack scenarios where the data have been intentionally altered in order to measure its effectiveness and to be able to propose possible evolutions that make it robust in these situations.

## Figures and Tables

**Figure 1 sensors-24-00479-f001:**
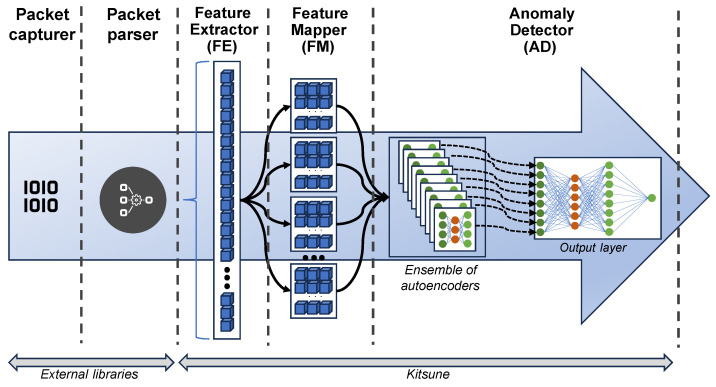
Kitsune’s architecture [9].

**Figure 2 sensors-24-00479-f002:**
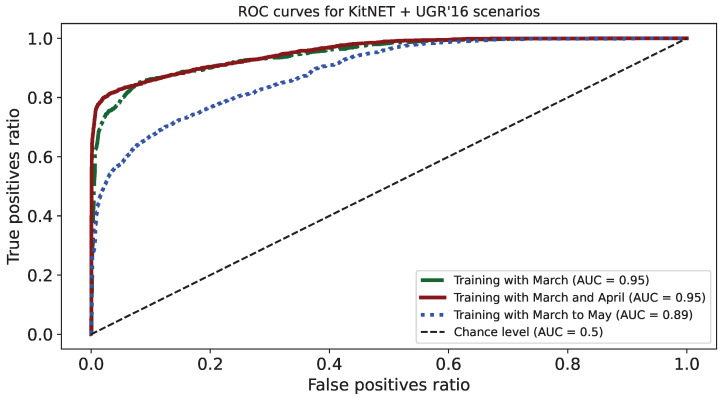
ROC curves corresponding to the experimental scenarios [20].

**Figure 3 sensors-24-00479-f003:**
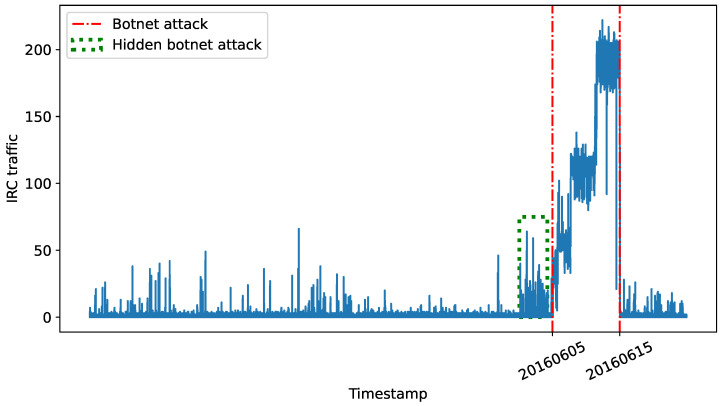
IRC traffic (blue line) registered in the UGR’16 calibration set.

**Figure 4 sensors-24-00479-f004:**
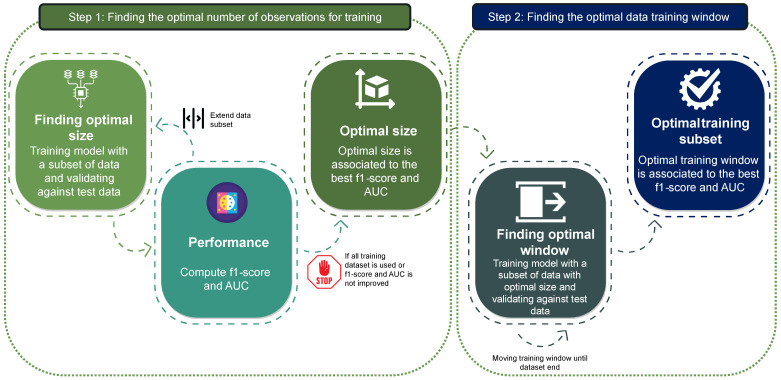
Proposed methodology workflow.

**Figure 5 sensors-24-00479-f005:**
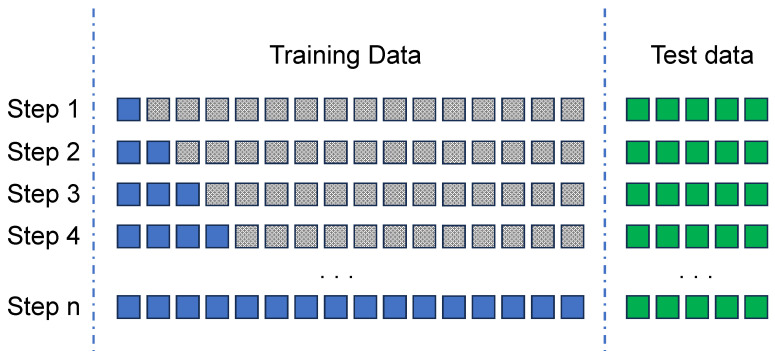
Finding the best size for the training window. Considering blue blocks as training data; grey ones as not used data and green blocks for validation.

**Figure 6 sensors-24-00479-f006:**
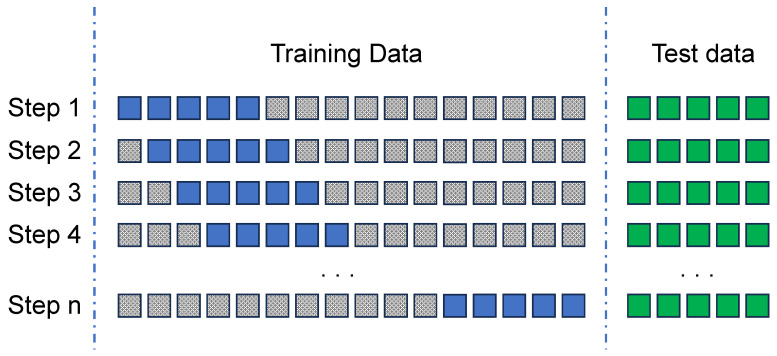
Step 2: Finding the optimal data training window. Considering blue blocks as training data; grey ones as not used data and green blocks for validation.

**Figure 7 sensors-24-00479-f007:**
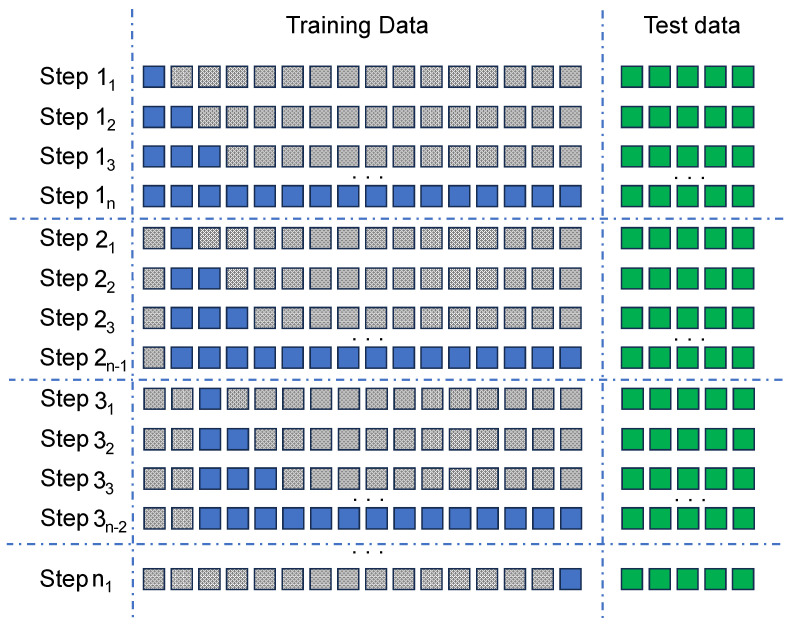
Exhaustive searching for the optimal data training window. Considering blue blocks as training data; grey ones as not used data and green blocks for validation.

**Figure 8 sensors-24-00479-f008:**
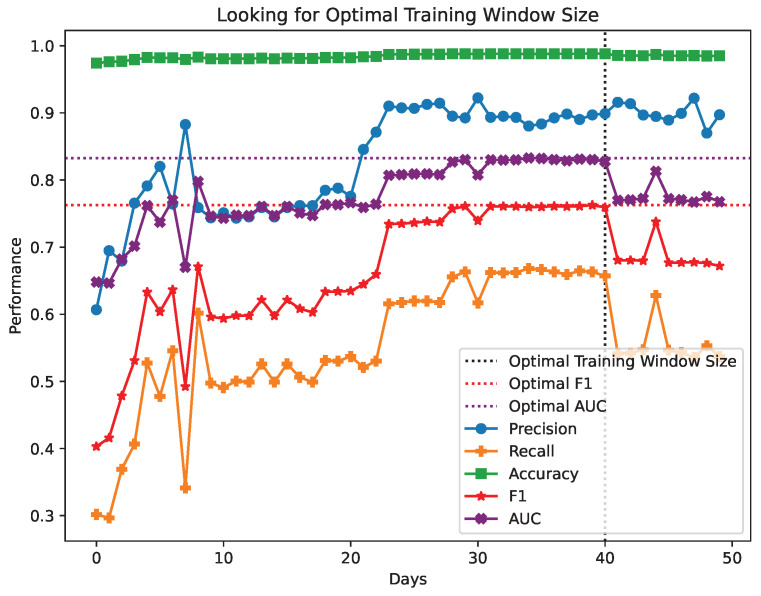
Performance metrics for iterations looking for the training window size.

**Figure 9 sensors-24-00479-f009:**
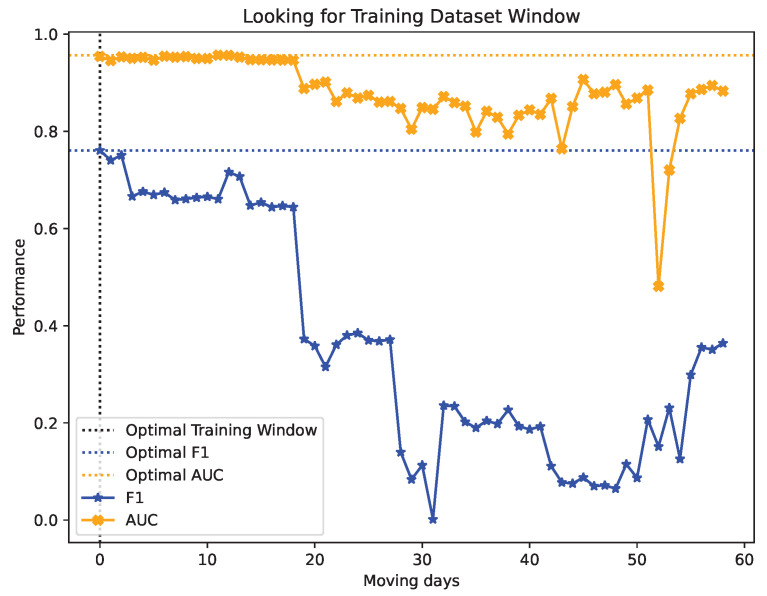
Performance metrics for iterations looking for the training window size.

**Figure 10 sensors-24-00479-f010:**
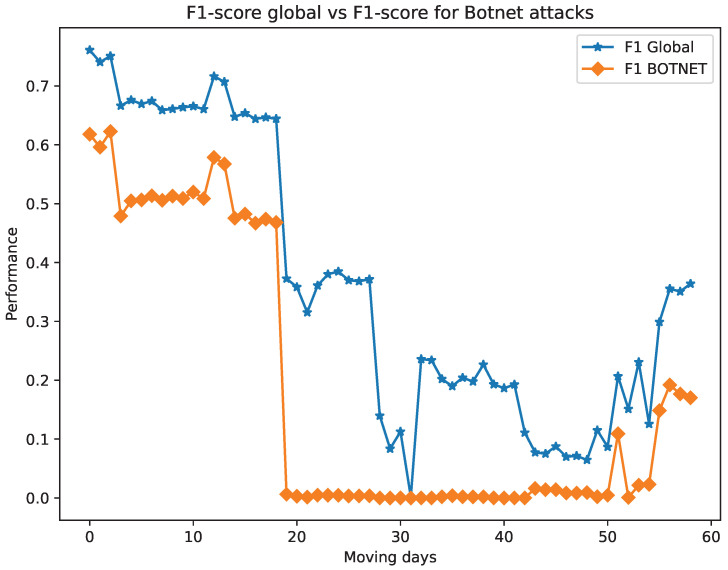
F1-score global vs F1-score for Botnet attacks.

**Table 1 sensors-24-00479-t001:** Overview of available network datasets.

Dataset	Availability	Collected Data	Labeled	Type	Duration *	Size **	Year	Freshness	Balanced
DARPA [26]	Public	packets	yes	synthetic	7 weeks	6.5TB	1998–1999	questioned	no
NSL-KDD [27]	Public	features	yes	synthetic	N.S.	5M o.	1998–1999	questioned	yes
Kyoto 2006+ [28]	Public	features	yes	real	9 years	93M o.	2006–2015	yes	yes
Botnet [29]	Public	packets	yes	synthetic	N.S.	14GB p.	2010–2014	yes	yes
UNSW-NB15 [30]	Public	features	yes	synthetic	31 hours	2.5M o.	2015	yes	no
UGR’16 [31]	Public	flows	yes	real	6 months	17B f.	2016	yes	no
CICIDS2017 [32]	Protected	flows	yes	synthetic	5 days	3.1M f.	2017	yes	no
IDS2018 [33]	Protected	features	yes	synthetic	10 days	1M o.	2018	yes	no
NF-UQ-NIDS [34]	Public	flows	yes	synthetic	N.S.	12M f.	2021	yes	no

* N.S. means not specified. ** Expressed in flows (f.), observations (o.), or packets (p.). An observation denotes a data point with all specified features.

**Table 2 sensors-24-00479-t002:** Detection results of applying KitNET over UGR’16 [20].

Scenario	Training Data *	Class	Precision	Recall	F1-Measure	Accuracy
Scenario 1	March	Normal	0.99	1	0.99	0.97
		Anomalous	0.75	0.17	0.27	
Scenario 2	March to April	Normal	0.98	1	0.99	0.99
		Anomalous	0.9	0.66	0.76	
Scenario 3	March to May	Normal	0.98	1	0.99	0.98
		Anomalous	0.87	0.23	0.36	

* Months considered as training (Mar. for March, Apr. for April).

**Table 3 sensors-24-00479-t003:** Detection ratios by attack type of applying KitNET over UGR’16 [20].

Attack	Scenario 1	Scenario 2	Scenario 3
DOS	34%	64%	56%
SCAN11	1%	1%	1%
SCAN44	51%	72%	57%
BOTNET	4%	74%	1%
UDPSCAN	0%	0%	0%
Total	16%	65%	22%

## Data Availability

UGR’16 dataset is available in https://nesg.ugr.es/nesg-ugr16/ (accessed on 20 June 2023), and KitNET is available in https://github.com/ymirsky/KitNET-py (accessed on 20 June 2023).

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
