# Peer review of "Methodology for the Detection of Contaminated Training Datasets for Machine Learning-Based Network Intrusion-Detection Systems†"

_sensors, 2024, doi:10.3390/s24020479_

Round 1

Reviewer 1 Report

Comments and Suggestions for Authors

The paper is well-written, easy to follow and I really appreciate that, of good contribution as well. However, I have the following main comments:

1) Why Kitsune architecture? Is it the best in that context, especially with the evolution of RNNs and their compatibility with time series-structured data?

2) Optimizing, disjointly, the number of observations and data window is suboptimal. A joint search/optimization should be done.

3) For the final results, the work lacks benchmarking with the state-of-the-art or some baselines.

Reviewer 2 Report

Comments and Suggestions for Authors

1.     The abstract fails to emphasize the main contributions of the paper adequately, including the proposed methodology and the addressed issues. It lacks clarity on what method was introduced and what problems were addressed, particularly in relation to the detection of labeling errors in the dataset.

2.     Is it necessary to extensively introduce datasets that were not utilized in your research in the second section on related work?

3.     The title of the 2.2 subsection, "Datasets Quality," is overly generic. It would be beneficial to provide a more specific and informative title that accurately reflects the content.

Comments on the Quality of English Language

1.     The paper contains sentences that deviate from the scholarly writing style and some instances of overly complex English expressions. It is recommended to make revisions for better adherence to academic writing conventions and to simplify intricate language constructions.

Reviewer 3 Report

Comments and Suggestions for Authors

This paper deals with the quality of network traffic datasets for Network Intrusion Detection Systems (NIDS) using machine learning. More precisely authors address the problem of contaminated and mislabeled training dataset. The main contribution of this paper is the proposition of a method to choose the optimal dataset for the training step. This is achieved by optimizing ROC AUC and F1 score. The proposed method is validated using UGR’16 dataset and Kitsune. The paper is well written in a clear English. It provides a complete state of the art about NIDS and network traffic datasets. However, the results seem to be closely dependent on the considered dataset. Authors should validate their proposition on several other datasets to prove the genericity of their proposition. The use of the sliding window is not clear: why should the optimal dataset be consecutive in time and not several days sampled among the whole dataset? Time and space complexity of this additional step (choosing the optimal training dataset) should be evaluated. Moreover, the use of Kitsune is not enough motivated.

Round 2

Reviewer 1 Report

Comments and Suggestions for Authors

I appreciate the quality of the paper from the beginning and, furthermore, the revised version. However, as selecting the training dataset is the paper's main contribution, the disjoint optimization approach for the number of observations and data window is suboptimal and thus cannot be accepted.

Reviewer 3 Report

Comments and Suggestions for Authors

I thank the authors for their reply and accept the current version of the paper.
